# Psychometric Properties and Measurement Invariance by Gender of the Abbreviated Three-Item Version of the Satisfaction with Life Scale in a Colombian Sample

**DOI:** 10.3390/ijerph19052595

**Published:** 2022-02-23

**Authors:** Begoña Espejo, Marta Martín-Carbonell, Irene Checa

**Affiliations:** 1Department of Behavioral Sciences Methodology, University of Valencia, 46010 Valencia, Spain; irene.checa@uv.es; 2Psychology Department, Cooperative University of Colombia, Santa Marta 470002, Colombia; martha.martinc@campusucc.edu.co

**Keywords:** Satisfaction with Life Scale, wellbeing, structural equation modeling, psychometric properties, measurement invariance, confirmatory factor analysis, health, quality of life, psychological assessment, Colombian population

## Abstract

(1) Background: The need to offer brief scales with items that can be answered with few response options is increasingly important in order to be able to access a broad range of the population. The three-item version of Diener’s Satisfaction with Life Scale has recently been proposed. The objective of this study is to study the psychometric properties of the three-item version of this Scale with five response options, as well as the measurement invariance by gender, in a Colombian sample; (2) Methods: A confirmatory factor model of the three items of the scale together with the Flourishing Scale has been tested, and the measurement invariance by gender of the model has been studied. The results offer a very satisfactory fit of the model, showing good evidence of construct and criterion validity, good indicators of reliability and measurement invariance by gender; (4) Conclusions: The three-item version of the Life Satisfaction Scale, previously adapted to the Colombian population with five response options, is presented as a valid and reliable measurement tool. In future studies, it would be convenient to study the test–retest reliability, as well as its psychometric properties in different samples and at a cross-cultural level.

## 1. Introduction

Research shows that the concept of subjective well-being is multidimensional [1]. In order to understand the psychological processes that underlie people’s search for happiness, positive psychologists developed different constructs to operationalize happiness and personal well-being. These constructs are usually categorized into the following two different research perspectives: hedonic and eudemonic. The hedonic perspective, characterized by subjective well-being, conceives of individual well-being as the evaluation that people make of the satisfaction they experience in their lives (cognitive component) and the balance between positive and negative affects (affective-emotional component) [2,3]. To assess the cognitive component, Diener and colleagues developed the Satisfaction with Life Scale (SWLS) [4]. In fact, subsequent studies have shown that the SWLS is related to the frequency of positive personal experiences, not their intensity [5]. To assess this intensity, which is the affective-emotional component of experiences, the Scale of Positive and Negative Experience (SPANE) was developed [6]. On the other hand, psychological well-being represents the eudemonic perspective, associating the well-being of the person with the development of their full potential as a human being. From this perspective, happiness is conceived of as a state in which people tend towards a purpose in life based on personal development and growth. To assess this well-being component, Diener and colleagues developed the Flourishing Scale (FS) [6].

According to its authors, the SWLS allows people to assess how satisfied they are with their life regardless of their emotional state at the time of responding, since it focuses more on the positive part of personal experiences. In fact, different studies have shown that high scores in satisfaction with life show direct relationships with variables related to health, and inverse relationships with variables related to emotional experiences that are perceived as negative [7,8]. This scale has also been adapted and its psychometric properties have been evaluated in a large number of countries and across different populations, including adolescents, adults, the elderly and clinical samples [9], always confirming its unifactorial structure. In this sense, it has been found that it presents statistically significant correlations with physiological measures [10,11], not only with other self-report measures, which is another guarantee of its construct validity. It has also been found that it is a scale capable of detecting differences in life satisfaction between groups of people with different circumstances in their lives, which would be an indicator of its differential validity [12]. Likewise, its predictive value has been verified, since it has been shown to be useful even for detecting changes in the scores associated with important life situations, such as for detecting the risk of possible suicidal behaviors [13,14]. Furthermore, measurement invariance across gender within the same culture has been verified in several studies [9,15,16].

Since its introduction, it has been used in hundreds of studies to assess life satisfaction [12,17] and has been adapted into many languages, studying psychometric properties in many countries around the world [17]. In the same way that its one-dimensional structure has been confirmed in the different adaptations carried out in multiple countries, cross-cultural studies have revealed the existence of important differences in the satisfaction with life that people from different countries show [9,18]. Although it could be thought that the differences are due to cultural and economic issues [18], some authors have subsequently pointed out that it may be due to how satisfaction with life is defined in different countries, to the conceptualization of the construct. In fact, in a study that collected data over the last 30 years on the SWLS, both men and women, within each country, were found to perceive this concept in the same way. However, the perception is different between age groups and between countries. In other words, it seems that there is measurement invariance by gender (within each country), but not by age groups or by country [9]. Likewise, in another study carried out the same year, the invariance between 26 countries was examined using three different methods (multigroup confirmatory factor analysis, maximum likelihood, and alignment optimization), consistently finding that there was a configural and metric invariance but no scalar invariance. However, items 1 and 3 did show scalar invariance [19]. Recently, measurement invariance of the SWLS has been studied among adolescents in 24 countries and regions [20]. The authors found three items on the scale (1, 2, and 4) that showed evidence of non-invariance across cultures. It was found that the measurement model works in a similar way depending on gender and age, and they suggested that the reason could be that the meaning of “ideal life” (item 1), “living conditions” (item 2) and “important things” (item 4) may have different meanings across cultures.

This reinforces what was stated by Pavot and Diener [17], who already pointed out a decade earlier that the fifth item of the scale was very different from the rest both in its content and in its psychometric qualities. For this reason, they recommended that if a researcher were particularly interested in assessing a respondent’s satisfaction with his or her current life, the data could be examined by discarding the fifth item.

Recently, Kjell and Diener [21] proposed a three-item version, based on the fact that from a psychometric perspective, the first three items of the SWLS have shown the highest factor weights and item-scale correlations. Thus, they described a study in which they used three datasets including test–retest data (N = 787; N = 860; N = 343), finding a very high internal consistency for the abbreviated three-item version, a solid test–retest reliability, very good fit indices, and measurement invariance across time and gender. Additionally, the scale demonstrated similar validity to the five-item one by producing similar correlations with other measures of well-being, mental health problems, and social desirability. These authors supported their proposal to condense the SWLS due to the importance of having brief measures that can be used in different contexts and populations, including online studies, longitudinal designs, screening studies, and others [21,22].

On the other hand, some authors recommend making modifications to the five-item SWLS response options to facilitate its application in different cultural contexts. The original response scale proposed by Diener and colleagues is a 7-point Likert-type scale. However, previous studies have shown that offering too many response options can be problematic for people with a low educational level [23]. Likewise, many response options can lead to confusion and boredom in the people who respond, and can also lead to difficulty in distinguishing small differences between the verbal anchors of the response categories [24]. In a cross-cultural measurement invariance study of the SWLS using Italian and African samples, it was found that the 7-point response scale was not sensitive to detecting low levels of life satisfaction, and they recommended using fewer response options, especially for the South African population [25]. Several studies in Spanish-speaking populations have used the 5-item SWLS with five response options instead of the seven in the original scale, both in Spain and in other Latin American countries [26,27,28,29,30,31,32,33], always showing good psychometric properties. Furthermore, we recently found excellent psychometric properties in the five-choice version of the SWLS in a large sample of Colombians [31].

So, to facilitate the applicability of a test, it must be easy to answer the questionnaires, so that people who have certain limitations in understanding the items are able to access it, as occurs with people with a low educational level or people with cognitive problems [34]. Thus, it is important to have a brief and easy-to-understand instrument, which is also valid for knowing how people evaluate their satisfaction with life. For this reason, the present study aims to evaluate, for the first time, the psychometric properties of the abbreviated version of Diener’s Satisfaction with Life Scale (version of 3 items and 5 response options) in the Colombian population.

## 2. Materials and Methods

### 2.1. Procedure and Participants

Sampling was non-probabilistic. Data were collected between August 2019 and February 2020. Survey data were collected online using the LimeSurvey program. Participants were recruited by email and on different social networks. An explanation of the study and a link to the platform were initially included. Participants were required to read and provide online informed consent before responding to the survey. The survey was answered anonymously, and no participant could be identified through their responses. Additionally, no one received compensation of any kind for participating. There were no missing data in the questionnaires used in this study. The program used makes it possible to establish the obligation to answer the questions. Therefore, people who at some point leave an item unanswered can no longer continue. On the other hand, the ethics committee of the Cooperative University of Colombia supervised and approved the planning and data collection. The sample consists of 1222 participants who answered both questionnaires in full, with a mean age of 25.66 years (SD = 8.66, Minimum 18, Maximum 67), and 64.4% were female. Most participants have completed university studies (42.9%) and many have completed high school (41.2%). Only 13% completed secondary school, while 2.9% completed or partially completed primary education. Most of the sample are single (75%), 22% are married or have an intimate partner, and 2.5% are divorced or widowed. Regarding main activity, 43.9% are full-time students, 23.5% are employed or self-employed, and the rest of the sample are unemployed, inactive or retired people (6.3%).

### 2.2. Measures

The Flourishing Scale (FS) [6] is a questionnaire made up of eight items that measure social–psychological prosperity and evaluate positive relationships, feelings of competence, as well as meaning and purpose in life. It uses a Likert-type agreement response scale that ranges between 1-totally disagree, and 7-totally agree, so that the higher the score, the better the perception people have of themselves. The version used in this study is the one adapted for the Colombian population [35]. Chronbach’s alpha in this sample is 0.916.

The Satisfaction with Life Scale (SWLS) [4] is a general five-item measure of satisfaction with the quality of life perceived. The original version uses a 7-point Likert-type scale that ranges from 1-strongly disagree to 7-strongly agree. To carry out this study, the version adapted for the Colombian population with 5 response options has been used [31], based on the version for the Spanish population [26]. In this version, the response scale ranges between 1-strongly disagree, and 5-strongly agree, so that the higher the score, greater satisfaction with life is perceived. In this sample, Chronbach’s alpha for the five items is 0.842, and 0.796 for the three-item version used to carry out this study. The items used to carry out the present study are “1. In most ways, my life is close to my ideal”, “2. I am satisfied with my life” and “3. The conditions of my life are excellent”.

### 2.3. Data Analyses

To study the factor structure of the SWLS three-item version, as it is a three-element version, the FS has been used for the following two purposes: on the one hand, to anchor the Satisfaction construct to the Flourishing construct in order to have degrees of freedom, and on the other hand, to obtain evidence of convergent validity of the abbreviated version of the SWLS. Since Flourishing is a component of subjective well-being, a positive and statistically significant correlation with life satisfaction is expected [6]. So, a structural equation model was estimated to test the model. The one-factor model has also been estimated, assuming that all the items (3 of the SWLS and 8 of the FS) measure a single factor of subjective well-being, to check whether both constructs can be considered one or if it is better to differentiate them.

To evaluate the fit of the model, χ^2^ and the following indices were used: the Comparative Fit Index (CFI) and the Tucker–Lewis Index (TLI) (in both cases, values above 0.90 indicate acceptable model fit, and above 0.95 indicate good fit to the model), the root-mean-square error of approximation (RMSEA) and Standardized Root-Mean-Square Residual (SRMR). A cutoff value close to 0.08 for SRMR and a cutoff value close to 0.06 for RMSEA are required to conclude that there is a relatively good fit [36]; values from 0.08 or higher indicate bad model fit [37,38]. To study reliability of the three-item SWLS, corrected item-total polyserial correlations (corrected homogeneity indexes) for the three items of the SWLS were calculated, as well as the Composite Reliability index (CR) and the Average Variance Extracted index (AVE). Values above 0.70 for CR and AVE are considered good. For AVE, values of 0.50 are considered acceptable [39]. Gender-based measurement invariance was also studied, evaluated by calculating three nested invariance models with the following successive restrictions: configural, metric and scalar. With a sample size greater than 300, a change of less than 0.010 in CFI, together with a change of less than 0.015 in RMSEA or a change of less than 0.030 in SRMR, would indicate that there is invariance [40].

An Mplus 8.7 [41] with Maximum Likelihood Robust (MLR) estimation has been used to carry out the CFA of the model and to test measurement invariance. Although the response scale is ordinal, some authors indicate that MLR can be used when the data distribution is not normal, and if the number of response options is greater than four [42,43]. In this case, MLR offers an estimation of the parameters with little variation and with less biased standard error estimates. Likewise, MLR offers good estimates of the correlations between the factors [44]. Corrected item-total polyserial correlations were calculated with Mplus 8.7, too. Finally, to describe sociodemographic variables and obtain descriptive statistics of the items, IBM SPSS 26 was used.

## 3. Results

Table 1 shows the descriptive data of the items for both scales (three-item version of the SWLS and the Flourishing Scale). The corrected item-total polyserial correlations (corrected homogeneity indexes) for the three items of the SWLS were 0.669 (s.e. = 0.013) for item 1, 0.692 (s.e. = 0.010) for item 1, and 0.650 (s.e. = 0.013) for item 3.

The one-factor demonstrated a bad fit of (χ^2^(44) = 504.655, *p* < 0.001), the other fit indicators indicated that the model does not fit the data, given that CFI = 0.877, TLI = 0.846, RMSEA = 0.093 (90% CI for RMSEA = [0.085, 0.010]) and SRMR = 0.074. Although χ^2^ is statistically significant (χ^2^(43) = 130.558, *p* < 0.001), the other fit indicators indicate that the two-factor model fits the data very well, i.e., CFI = 0.977, TLI = 0.970, RMSEA = 0.041 (90% CI for RMSEA = [0.033, 0.049]) and SRMR = 0.032. All factor loadings in this model were statistically significant, with the standardized values ranging from 0.613 to 0.844 (*p* < 0.001 all of them) (see Figure 1). For the three-item SWLS version, the values for the CR and the AVE were 0.797 and 0.567, respectively. For the FS, the values for the CR and the AVE were 0.929 and 0.624, respectively. The correlation between the latent variables was 0.597 (*p* < 0.001), which also offers good evidence of convergent validity.

The results of the measurement invariance models by gender are shown in Table 2. As can be seen, there is an excellent fit of the model for both genders. Likewise, the existence of configural, metric and scalar invariance is clearly observed. When comparing the values of the latent means of both groups, no differences by gender are observed in the scores of the three-item version of the SWLS (b = −0.018, z = −0.376, *p* = 0.707).

## 4. Discussion

The aim of this research was to study the psychometric properties and gender measurement invariance of the Satisfaction with Life Scale, in its abbreviated three-item version proposed by Kjell and Diener [21]. The results indicate that this reduced version of the scale shows good construct validity, as well as good criterion validity with the Flourishing Scale. Likewise, the reliability results indicate that the items show good values of homogeneity indices that were corrected for the three-item version, and the Composite Reliability value was good. Although the value offered by the AVE for the three-item version of the SWLS is not optimal, it is an acceptable one, which indicates that the three items of the abbreviated satisfaction scale explain a reasonable percentage of the variance.

Likewise, the proposed model of satisfaction and flourishing shows a strong invariance by gender, which allows the observed scores to be used to make comparisons between both groups. In addition, when comparing the means, no differences between men and women in the mean satisfaction with life were observed, calculated from the three items that were considered.

In addition, it must be considered that this reduced version of the satisfaction scale was used with a Likert-type response scale with five response options, previously validated in the Colombian context. As several authors propose, while brief measurement instruments can be more useful to avoid fatigue in the participants, especially in longitudinal studies [21,22], using a smaller number of response options can avoid errors in the interpretation of verbal anchors which are sometimes very subtle. These very small differences between the verbal anchors can lead to confusion and boredom in the people who respond, and can also lead to difficulty in distinguishing small differences between the verbal anchors of the response categories [24]. Other studies show that offering too many response options can be problematic for people with a low educational level, as well as for the elderly or people with disabilities [23]. As has been seen in another study, too many response options may not detect changes in levels of satisfaction with life; therefore, it is advisable to reduce the number of response options [25].

To date, the SWLS with five response options has already been used in the Spanish-speaking population, instead of the seven of the original scale [26,27,28,29,30,31,32,33], consistently showing good psychometric properties. The results of this study indicate that the reduced version of this scale made up of only three items, and with only five response options, has shown good psychometric properties. For this reason, its use can be very useful to carry out studies on the psychological well-being of people, being able to include people who are usually excluded from them in said studies.

Even so, this study has certain limitations that should be considered. The first is that the study sample is non-probabilistic, which limits the generalization of the results, taking into account that there is great cultural diversity in Colombia. This can be a problem when it comes to generalizing the results obtained in terms of satisfaction with life. In addition, young people predominate the sample because it was easier to access them, especially since the survey was offered online. It also happens that, for the most part, the sample of this study shows a high educational level, which is not representative of the Colombian population, in which a small percentage of people, but not a negligible number, cannot read or write [45]. Another limitation of the study is that, although invariance by gender was found in this case, the groups were not equal in size. In this sample, there were more women (about 64%) than men. If the groups are very unbalanced, measurement invariance violations may be overlooked [46]. Therefore, in future studies it would be advisable to obtain a larger sample of men and to test the invariance by gender again.

In the future, a more representative sample of the Colombian population should be surveyed, including people who live in rural areas and indigenous populations. It would also be appropriate to carry out studies of the temporal stability of the measure and comparative studies by age group and socioeconomic level. Likewise, studying the cross-cultural invariance would allow us to know if the satisfaction measures obtained with this index composed of three items shows measurement invariance by country.

## 5. Conclusions

The three-item measure of the Satisfaction with Life Scale has shown good psychometric properties in the Colombian study sample, as well as measurement invariance by gender. Due to its brevity, as well as the use of five response options, it is a very suitable instrument to obtain reliable and valid indicators of satisfaction with life in Colombia. Future studies should check the invariance of measurement between other types of Colombian populations and between countries to make comparisons between these groups.

## Figures and Tables

**Figure 1 ijerph-19-02595-f001:**
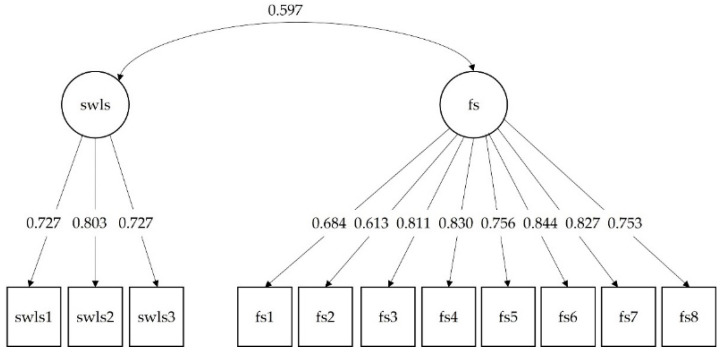
Standardized values for the structural equation model of the 3-item version of the Satisfaction with Life Scale (SWLS) and the Flourishing Scale (FS). Note: swls = Satisfaction with Life Scale; fs = Flourishing Scale; swls1 = In most ways, my life is close to my ideal; swls2 = I am satisfied with my life; swls3 = The conditions of my life are excellent. All coefficients are statistically significant (*p* < 0.001).

**Table 1 ijerph-19-02595-t001:** Descriptive statistics for the items of the 3-item version of the Satisfaction with Life Scale (SWLS) and the Flourishing Scale (FS).

	Mean	Standard Deviation	Skewness	Kurtosis
SWLS1	3.67	1.025	−0.747	0.035
SWLS2	4.00	1.016	−0.945	0.353
SWLS3	4.05	0.899	−1.031	1.140
FS1	6.08	1.491	−2.116	4.095
FS2	5.40	1.458	−1.164	1.056
FS3	5.84	1.385	−1.765	3.180
FS4	6.09	1.303	−2.199	5.336
FS5	5.69	1.282	−1.518	2.762
FS6	5.93	1.249	−1.802	3.921
FS7	6.00	1.385	−1.796	3.177
FS8	5.77	1.261	−1.458	2.511

Note: SWLS1 = In most ways, my life is close to my ideal; SWLS2 = I am satisfied with my life; SWLS3 = The conditions of my life are excellent.

**Table 2 ijerph-19-02595-t002:** Measurement invariance models of the 3-item version of the SWLS Flourishing Scale by gender (reference group: men).

Model	χ²	df	Δχ²	Δgl	CFI	RMSEA	SRMR	ΔCFI	ΔRMSEA	ΔSRMR
Men	50.774 *	43			0.994	0.020	0.030			
Women	119.226 *	43			0.970	0.047	0.035			
Configural	172.869 *	86	-	-	0.977	0.041	0.034	-	-	-
Metric	187.198 *	95	13.430	9	0.976	0.040	0.047	−0.001	−0.001	0.013
Scalar	199.228 *	104	8.286	9	0.975	0.039	0.046	−0.001	−0.001	−0.001

Note: df = degrees of freedom; Δχ² = Chi Square increase; Δgl = increase in degrees of freedom; CFI = comparative fit index; RMSEA = Root-Mean-Square error of approximation; SRMR = Standardized Root Mean Square Residual; ΔCFI = CFI change; ΔRMSEA = RMSEA change; ΔSRMR = SRMR change. * *p* < 0.001.

## Data Availability

Data can be provided upon request.

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
