# Peer review of "Psychometric Properties and Measurement Invariance by Gender of the Abbreviated Three-Item Version of the Satisfaction with Life Scale in a Colombian Sample"

_ijerph, 2022, doi:10.3390/ijerph19052595_

Round 1

Reviewer 1 Report

Literature Review

  1. Citation is needed for some statements in lines 57-59, 65-66, and 67-69.
  2. There is a typo error in Line 97-98, "a solid test-retest reliability retest"
  3. Lines 104-118, specify whether the SWLS refers to the original 5-item or the 3-item version.
  4. Lines 119-126, although the authors have reviewed the limitations of the SWLS in previous paragraphs, it is necessary to explicitly indicate the problem statement of the study here. It is helpful to indicate whether the 3-item version of the SWLS has been tested in a Colombian sample.

Materials and Methods

  1. Kindly indicate if the dataset consists of any missing values, the data collection period, and ethical clearance information.
  2. Kindly indicate if the original English versions of the SWLS and Flourishing Scale (FS) were used in the study.
  3. lines 156-157, "Chronbach’s alpha for the five items is 0.842". Which version of the SWLS was used in the study, 5-item or 3-item? If the 5-item version was used, the authors shall examine and report the psychometric properties of the 5-item version for comparison purposes.
  4. Line 165, the statement "Flourishing is a component of subjective well-being" requires justification and citation. This is because the authors reported that the FS measures dimensions such as positive relationships, feelings of competence, and meaning and purpose in life that belong to eudemonic well-being. Therefore, the use of FS is questionable.
  5. Lines 171-172, "values between .06 and .08 indicate acceptable model fit, and values below .05 indicate good fit" What about values between .05 and .059?
  6. Line 176, "In both cases, values above 0.70 are considered good." The writing is misleading because it is not what the cases refer to.
  7. Explain the types of validity and how they were tested in the study.
  8. Line 180, " less than −0.010" is misleading. It should be the change in value (or absolute value) less 0.010. 
  9. Would it be possible to test the age-based measurement invariance with the large sample size?    

Results

  1. The authors only tested the two-factor model. Other competing models such as the one-factor model shall be tested to indicate if the two-factor model is superior to other models.
  2. Line 206, "The values for the CR and the AVE were 0.797 and 0.567". Kindly specify whether they are the values for the SWLS, FS, or the SEM model. The authors are suggested to report the values for both scales.
  3. Are the factor loadings shown in Figure 1 the standardized values?
  4. Line 225,  ΔCFI refers to the change (not limited to increase) in CFI value.

Discussion

  1. There is a large difference in the sample size of the two gender groups. The authors shall discuss the possibility that the measurement invariance results could be influenced by the difference.

Reviewer 2 Report

Thank you for the opportunity to review the manuscript entitled "Psychometric properties and measurement invariance by Gender of the abbreviated three-item version of the Satisfaction With Life Scale in a Colombian sample". This is an excellent study that rigorously presents the validation of short SWLS in the Colombian population. The Introduction presents current scientific literature clearly and comprehensively, which is a base for presentation the aims of this study. The Methods section is well-written and allows for replication of the study. However, it is suggested to add information about how many people refused to participate in the study, and what was the response rate. How many missing values were in the study, and how the authors coped with this issue? What was statistical power? Results are transparent and clearly shown, using tables and figure presenting a CFI model. However, the note of Figure 1 is inconsistent with the data presented on the picture (swls3 and swls5 are on the picture, while swls2 and swls3 are in the note). Please be consistent to avoid misleading. Both the discussion and the conclusions are related and supported by the results. Summarizing, this is a well-written manuscript. I really enjoyed reading this paper.

Author Response

Thank you very much for your words.

Regarding the sample, data were collected between August 2019 and February 2020. There are no missing data in the questionnaires used in this study. The program used makes it possible to establish the obligation to answer the questions. Therefore, people who at some point leave an item unanswered can no longer continue. The final sample used for this study consisted of 1222 participants who answered both questionnaires in full. There are no missing values, then. On the other hand, the ethics committee of the Universidad Cooperativa de Colombia supervised and approved the planning and data collection. In addition, as indicated in the description of the sample, before starting the survey, information was provided about the data protection policy and the participants had to accept informed consent before continuing, otherwise the survey was terminated. This information has been included in this section.

Regarding the Figure 1, you are right. It was a mistake. As the order of the items in our sample was different, the number of these items in the figure are different. We have corrected them in the figure. The footnote is the same, then.

Thank you very much again for your words and for detecting the errors in the figure.

Round 2

Reviewer 1 Report

I thank the authors for their efforts in further improving the quality of the manuscript. The revised version is satisfactory.

Below are further suggestions for the authors' consideration:

  1. Lines 179-181, "the root-mean-square error of approximation (RMSEA) and Standardized Root-Mean Square Residual (SRMR) (values between .06 and .08 indicate acceptable model fit, and values below 0.05 indicate good fit)". Please verify whether the cut-off for good fit model is 0.05 and below (i.e., the value 0.05 is included).
  2.  Lines 239-240, "ΔCFI = CFI increase; ΔRMSEA = RMSEA increase; ΔSRMR = SRMR increase". The delta symbol (Δ) means “change” or “the change” in maths. To precisely present the information, it is suggested to replace "CFI increase" to "change/different in CFI". The same applies to other indicators. 
  3. It is believed that some readers may have the same inquiry of why age-based measurement invariance was not tested. It would be great if the authors can clarify this in the Data Analyses section. 

Author Response

1. This sentence has been rewritten, with new references:

A cutoff value close to 0.08 for SRMR and a cutoff value close to 0.06 for RMSEA are needed to conclude that there is a relatively good fit (Hu and Bentler 1999); values from 0.08 indicate bad model fit (Hu and Bentler 1995; Marsh, Balla, and Hau 1996).

2. The word has been changed

3. This information has been provided in a new paragraph (lines 192-195):

The invariance of measurement by age could not be calculated since most of the sample is young (75% of the participants are 26 years old or younger). For this reason, it has not been possible to form groups of different ages.

The new changes are highlighted in green.